# REVISITING DOMAIN RANDOMIZATION VIA RELAXED STATE-ADVERSARIAL POLICY OPTIMIZATION

## ABSTRACT

Domain randomization (DR) is widely used in reinforcement learning (RL) to bridge the gap between simulation and reality through maximizing its *average returns* under the perturbation of environmental parameters. Although effective, the methods have two limitations: (1) Even the most complex simulators cannot capture all details in reality due to finite domain parameters and simplified physical models. (2) Previous methods often assume that the distribution of domain parameters is a specific family of probability functions, such as a normal or a uniform distribution, which may not be correct. To enable robust RL via DR without the aforementioned limitations, we rethink DR from the perspective of *adversarial state perturbation*, without the need for re-configuring the simulator or relying on prior knowledge about the environment. We point out that perturbing agents to the worst states during training is naïve and could make the agents *over-conservative*. Hence, we present a *Relaxed State-Adversarial Algorithm* to tackle the over-conservatism issue by simultaneously maximizing the average-case and worst-case performance of policies. We compared our method to the state-of-the-art methods for evaluation. Experimental results and theoretical proofs verified the effectiveness of our method.

## 1 INTRODUCTION

Most reinforcement learning (RL) agents are trained in simulated environments due to the difficulties of collecting data in real environments. However, the domain shift, where the simulated and real environments are different, could significantly reduce the agents' performance. To bridge this *"reality gap"*, domain randomization (DR) methods perturb environmental parameters (Tobin et al., 2017; Rajeswaran et al., 2016; Jiang et al., 2021), such as the mass or the friction coefficient, to simulate the uncertainty in state transition probabilities and expect the agents to maximize the return over the perturbed environments. Despite its wide applicability, DR suffers from two practical limitations: (i) DR requires direct access to the underlying parameters of the simulator, and this could be infeasible if only off-the-shelf simulation platforms are available. (ii) To enable sampling of environmental parameters, DR requires a prior distribution over the feasible environmental parameters. However, the design of such a prior typically relies on domain knowledge and could significantly affect the performance in real environments.

To enable robust RL via DR without the above limitations, we rethink DR from the perspective of *adversarial state perturbation*, without the need for re-configuring the simulator or relying on prior knowledge about the environment. The idea is that perturbing the transition probabilities can be equivalently achieved by imposing perturbations upon the states after nominal state transitions. To substantiate the idea of state perturbations, a simple and generic approach from the robust optimization literature (Ben-Tal & Nemirovski, 1998) is taking a *worst-case* viewpoint and perturbing the states to nearby states that have the lowest long-term expected return under the current policy (Kuang et al., 2021). While being a natural solution, such a worst-case strategy could suffer from severe *over-conservatism*. We identify that the over-conservative behavior results from the tight coupling between the need for temporal difference (TD) learning in robust RL and the worst-case operation of state perturbation. Specifically: (1) In robust RL, the value functions are learned with the help of bootstrapping in TD methods since finding nearby worst-case states via Monte-Carlo sampling is NP-hard (Ho et al., 2018; Chow et al., 2015; Behzadian et al., 2021). (2) Under the worst-case state perturbations, TD methods would update the value function based on the local minimum within a neighborhood of the nominal next state and is, therefore, completely unaware of the value of the

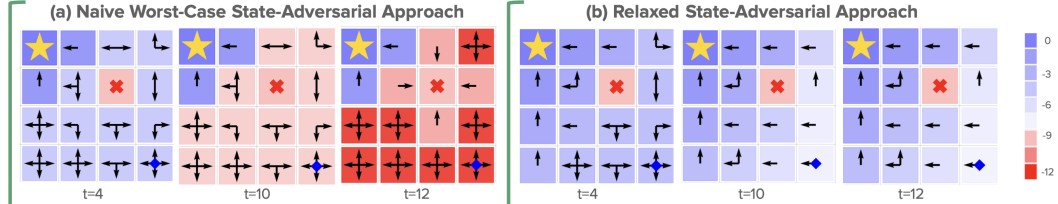

Figure 1: We illustrate the over-conservative issue of the naive worst-case state-adversarial policy optimization using a $4 \times 4$ shortest-path grid world environment. The star, cross, and dot represent the goal, the trap, and the initial state, respectively. The terminal rewards of the trap and the goal are $-10$ and $0$. We use an arrow to represent the action that has the highest value at each state. Multiple arrows in a state indicate that the actions have equal Q-values. We also use the color to indicate the value of the best action at each state. In (a), the agent trained by the naïve worst-case state-adversarial approach fails to learn how to reach the goal. What's worse, under TD updates, the worst-case state perturbation makes the trap state indistinguishable from other states. As a result, the agent ultimately learns to move toward the trap state after 12 training iterations. In (b), our relaxed state-adversarial approach avoids the over-conservatism issue by considering both the average-case and worst-case environments. We refer readers to Appendix A.1 for the step-by-step evolution of the value functions.

nominal next state. As a result, the learner could fail to identify or explore those states with potentially high returns. To further illustrate this phenomenon, we consider a toy grid world example of finding the shortest path toward the goal, as shown in Figure 1(a). Although the goal state has a high value, the TD updates cannot propagate the value to other states since all nominal state transitions toward the goal state are perturbed away under the worst-case state-adversarial method. What's even worse, the agent ultimately learns to move toward the trap state due to the compounding effect of TD updates and worst-case state-adversarial perturbations. Notably, in addition to the grid world environment, such trap terminal states also commonly exist in various RL problems, such as the locomotion tasks in MuJoCo. As a result, there remains one critical unanswered question in robust RL: *how to fully unleash the power of the state-adversarial model in robustifying RL algorithms without suffering from over-conservatism?*

To answer this question, we introduce relaxed state-adversarial perturbations. Specifically: (1) Instead of taking a pure worst-case perspective, we simultaneously consider both the average-case and worst-case scenarios during training. By incorporating the average-case scenarios, the TD updates can successfully propagate the values of those potentially high-return states to other states and thereby prevent the over-conservative behavior (Figure 1(b)). (2) To substantiate the above idea, we introduce a relaxed state-adversarial transition kernel, where the average-case environment can be easily represented by the interpolation of the nominal and the worst-case environments. Under this new formulation of DR, each interpolation coefficient corresponds to a distribution of state adversaries. (3) Besides, based on this formulation, we theoretically quantify the performance gap between the average-case and the worst-case environments; and prove that maximizing the average-case performance can also benefit the worst-case performance. (4) Accordingly, we present Relaxed state-adversarial policy optimization, a bi-level framework that optimizes the rewards of the two cases alternatively and iteratively. One level updates the policy to maximize the average-case performance, and the other updates the interpolation coefficient of the relaxed state-adversarial transition kernel to increase the lower bound of the return of the worst-case environment.

## 2 RELATED WORK

**Robust Markov Decision Process (MDP) and Robust RL.** Robust MDP aims to maximize rewards in the worst situations if the testing environment deviates from the training environment (Nilim & El Ghaoui, 2005; Iyengar, 2005; Wiesemann et al., 2013). Due to the large searching space, the complexity of robust MDP grows rapidly when the dimensionality increases. Therefore, Tamar et al. (2014) developed an approximated dynamic programming to scale up the robust MDPs paradigm. Roy et al. (2017) extended the method to nonlinear estimation and guaranteed the convergence to a regional minimum. Afterward, the works of (Wang & Zou, 2021; Badrinath & Kalathil, 2021)

study the convergence rate when applying function approximations under assumptions. Derman et al. (2021) showed that the regularized MDPs are a particular instance of robust MDPs with uncertain rewards. They solved regularized MDPs rather than robust MDPs to reduce computation complexity. Grand-Clément & Kroer (2020) developed efficient proximal updates to solve the distributionally robust MDP via gradient descent and improved the convergence rate. However, although several approximations were presented, such model environments are still too restrictive, and they cannot be used to solve real-world problems.

**Adversary in Observations.** Even a small perturbation to observations may significantly degrade agents' performance because deep neural networks are vulnerable to inputs constructed by adversaries (Huang et al., 2017). Therefore, methods were presented to train agents under environments with adversarial attacks to improve their robustness (Kos & Song, 2017; Pattanaik et al., 2018). To guarantee a lower-bound performance, the works of (Lütjens et al., 2020; Wang et al., 2019) adopted the idea of certified defense used in classification problems. When making discrete actions, agents are certifiably robust to adversaries in observation within the $\epsilon$ distance ($L_p$-norm). Since most real-world problems are continuous, there were also methods (Weng et al., 2019; Zhang et al., 2020; Oikarinen et al., 2021; Zhang et al., 2021) presented to improve agents' robustness for continuous actions.

**Domain Randomization.** Environments can induce the uncertainty of transition probabilities. To simulate this circumstance, one can perturb the environmental parameters of a simulator to reasonably change transition probabilities when training agents (Huang et al., 2021; Tobin et al., 2017; Jiang et al., 2021; Igl et al., 2019; Cobbe et al., 2019). Specifically, Tobin et al. (2017) randomly sampled environmental variables and optimized the agents' average reward. Given that a significant perturbation may fail the training, Cobbe et al. (2019) increased the level of difficulty step by step when training agents to improve their average rewards. Jiang et al. (2021) further considered the expected return in the optimal case and introduced monotonic robust policy optimization to maximize the average-case and worst-case returns simultaneously. Since perturbing transition probabilities through environmental parameters demands prior knowledge, Kuang et al. (2021) transferred states to the nearby local minimum based on gradients obtained from the value function to imitate environmental disturbance. Igl et al. (2019) injected selective noise based on a variational information bottleneck and value networks to prevent models from overfitting the training environment. The regularization helps agents resist the uncertainty of state transition probabilities.

Our method perturbs states through the gradients of the value function, as Kuang et al. (2021) did. However, pushing states toward the nearby local minimum will make agents over-conservative because they consider only the worst-case scenarios. We present the relaxed state adversarial perturbation and optimize both the average-case and worst-case environments to overcome this problem.

## 3 PRELIMINARIES

A robust Markov decision process (robust MDP) is characterized by a tuple $(\mathcal{S}, \mathcal{A}, \mathcal{P}, R, \mu, \gamma)$, where $\mathcal{S}$ is the state space, $\mathcal{A}$ is action space, $\mathcal{P}$ is the *uncertainty set* that contains all possible transition kernels, $R : \mathcal{S} \times \mathcal{A} \to [-R_{\max}, R_{\max}]$ is the reward function, $\mu$ is the initial state distribution, and $\gamma \in (0, 1)$ is the discount factor. Let $P_0 \in \mathcal{P}$ denote the *nominal transition kernel*, which characterizes the transition dynamics of the nominal environment without perturbation. We define the total expected return under a policy $\pi$ and a transition kernel $P \in \mathcal{P}$ as

$$J(\pi|P) := \mathbb{E}_{s_0 \sim \mu, a_t \sim \pi(\cdot|s_t), s_{t+1} \sim P(\cdot|s_t, a_t)} \left[ \sum_{t=0}^{\infty} \gamma^t R(s_t, a_t) \right]. \tag{1}$$

For ease of exposition, we also define the value function under policy $\pi$ and transition kernel $P$ as $V_P^\pi(s) := \mathbb{E}_{a_t \sim \pi(\cdot|s_t), s_{t+1} \sim P(\cdot|s_t, a_t)} \left[ \sum_{a_t=0}^{\infty} \gamma^t R(s_t, a_t) | s_0 = s \right]$. To learn a policy in a robust MDP, the DR approaches are built on two major design principles: (1) *Construction of uncertainty set*: DR presumes that one could have access to the environment parameters of the simulator. The uncertainty set $\mathcal{P}$ is constructed by specifying the possible range of one or multiple environment parameters, typically based on some domain knowledge. (2) *Average-case perspective*: DR resorts to maximizing the average performance with respect to some pre-configured distribution $\mathcal{D}$ over the uncertainty set $\mathcal{P}$, i.e., $\mathbb{E}_{P \sim \mathcal{D}}[J(\pi|P)]$.

## 4    DOMAIN RANDOMIZATION VIA RELAXED STATE-ADVERSARY

### 4.1    CONNECTING DOMAIN RANDOMIZATION AND STATE PERTURBATION

Conventional DR methods enforce attacks on state transitions by perturbing the environment parameters of a simulator. This goal can be achieved by perturbing the state after each nominal transition (Kuang et al., 2021): Let $(s, a)$ be some state-action pair, and $\Gamma : \mathcal{S} \to \mathcal{S}$ be a state perturbation function. In a nominal environment, the probability of the transition to some state $s'$ under $s, a$ is $P(s'|s, a)$. Under the state perturbation $\Gamma$, the probability becomes $P(\Gamma(s')|s, a)$. However, this state adversarial attack is too effective since a value function considers the expected future return, and a perturbation to an early state may significantly influence the later states. The over-conservatism problem therefore occurs. We present a relaxed state-adversarial policy optimization to overcome the problem. We also prove that the relaxed MDP enjoys two main properties under relaxation: (1) it stands for the average performance of the uncertainty set; (2) it guarantees the improvement the performance of the worst-case MDP. Further, we prove that a specific average-case MDP corresponds to a relaxation parameter. Hence, we propose an algorithm for adapting the relaxation parameters during training.

### 4.2    STATE-ADVERSARIAL MDPS AND UNCERTAINTY SETS

State-adversarial attacks perturb the current states to neighboring states with the lowest values. This perturbation process can be captured by a state-adversarial transition kernel, which connects the nominal MDP and the resulting state-adversarial MDP. For ease of exposition, for each state $s \in \mathcal{S}$, we define $\mathcal{N}_\sigma(s) := \{s'|d(s, s') \leq \sigma\}$ to be the $\sigma$-neighborhood of $s$, where $d(s, s')$ can be any distance metric. In this study, we use $L_\infty$-norm.

**Definition 1** (State Perturbation Matrix). *Given a policy $\pi$ and a perturbation parameter $\sigma \geq 0$, the state perturbation matrix $Z_\sigma^\pi$ with respect to $\pi$ is defined as follows: for each pair of states $i, j \in \mathcal{S}$,*

$$Z_\sigma^\pi(i, j) := \begin{cases} 1, & \text{if } j = \arg\min_{s \in \mathcal{N}_\sigma(i)} V^\pi(s), \\ 0, & \text{otherwise.} \end{cases} \quad (2)$$

The justifications for choosing the above surrogate perturbation model are two-fold: (1) The model can be interpreted as constructing adversarial examples for the true states. (2) The perturbation model is closely related to the perturbation of environment parameters, which serve as the standard machinery in the canonical DR formulation, as described in (Kuang et al., 2021).

**Remark 1.** *In continuous state space, the* $\arg\min$ *in Equation 2 can be computed by adapting the fast gradient sign method (FGSM) (Goodfellow et al., 2014). Let $V$ be a value function (i.e., network) with parameter $\phi$, $s$ be a state, and $\epsilon$ be the strength of perturbation. FGSM finds the perturbed state $\Gamma(s) = s - \epsilon \cdot sign(\nabla_s V(\phi, s))$ that has the minimum value, where $||s - \Gamma(s)||_\infty \leq \epsilon$, and the gradient at $s$ is computed using back-propagation.*

**Definition 2** (State-Adversarial MDP). *For any policy $\pi$, the corresponding state-adversarial MDP with respect to $\pi$ is defined as a tuple $(\mathcal{S}, \mathcal{A}, P_\sigma^\pi, R, \mu, \gamma)$, where the state-adversarial transition kernel $P_\sigma^\pi$ is defined as*

$$P_\sigma^\pi(\cdot|s, a) := [Z_\sigma^\pi]^\top P_0(\cdot|s, a), \quad \forall(s, a) \in \mathcal{S} \times \mathcal{A}. \quad (3)$$

Recall that $P_0$ is the nominal transition kernel. We use the notation $P_\sigma^\pi = [Z_\sigma^\pi]^\top P_0$ in the later paragraphs for simplicity. Note that the state adversarial transition matrix $Z_\sigma^\pi$ depends on the strength of perturbation $\sigma$. Each perturbation radius $\sigma$ results in a unique state-adversarial MDP $P_\sigma^\pi$.

**Remark 2.** The state-adversarial MDP defined in Definition 2 involves perturbation of the true states, which is fundamentally different from the perturbation of observations (Zhang et al., 2020).

**Definition 3** (Uncertainty Set). *Given a radius $\epsilon > 0$, the uncertainty set induced by state-adversarial perturbations, denoted by $\mathcal{P}_\epsilon^\pi$, is defined as*

$$\mathcal{P}_\epsilon^\pi := \{P_\sigma^\pi : P_\sigma^\pi = [Z_\sigma^\pi]^\top P_0 \text{ and } \sigma \leq \epsilon\}. \quad (4)$$

The adversarial attack transits agents toward low-value states. Agents trained using this state adversarial MDP would prevent themselves from falling into the worst situation (Kuang et al., 2021). However, a large $\epsilon$ will make agents too conservative and fail to reach any goal state because its value cannot be propagated to neighboring states by the TD updates (Figure 1). Although using a small $\epsilon$ can ease the problem, agents would completely omit the risks outside the bounding area. Besides, this strategy is unachievable in a discrete environment due to the lower-bound value of $\epsilon$. For example, the agent's movement in the grid world is one hop and cannot be reduced.

**Lemma 1** (Monotonicity of Average Value in Perturbation Strength). *Under the setting of state adversarial MDP, the value of the local minimum monotonically decreases as the bounded radius $\sigma$ increases. Let $x$ be a positive real number. The reward function $J$ satisfies*

$$J(\pi|P_\sigma^\pi) \geq J(\pi|P_{\sigma+x}^\pi), \quad \forall\pi. \tag{5}$$

The proof is in Appendix A.3. Notably, Lemma 1 indicates that among the transition kernels in the uncertainty set $\mathcal{P}_\epsilon^\pi$, the worst-case occurs when $\sigma = \epsilon$.

### 4.3 RELAXED STATE-ADVERSARIAL MDPS

We present a relaxation framework to address the over-conservatism issue. To begin with, we consider a relaxation on the state-adversarial transition kernel as follows:

**Relaxed state-adversarial transition kernel.** Given $\epsilon > 0$ and $\alpha \in [0,1]$, the $\alpha$-relaxed state-adversarial transition kernel is defined as a convex combination of the nominal and the state-adversarial transition kernels, i.e.,

$$P_\epsilon^{\pi,\alpha}(\cdot|s,a) = \alpha P_0(\cdot|s,a) + (1-\alpha)P_\epsilon^\pi(\cdot|s,a). \tag{6}$$

**Connecting relaxed state-adversarial MDPs with domain randomization.** DR methods demand a prior distribution for computing the average case performance. Let $\mathcal{D}$ be a distribution over the uncertainty set $\mathcal{P}_\epsilon^\pi$. In the following, we show that applying DR with respect to $\mathcal{D}$ is equivalently cast optimizing an objective under a relaxed state-adversarial transition kernel.

**Lemma 2** (Relaxation parameter $\alpha$ as a prior distribution $\mathcal{D}$ in domain randomization). *For any distribution $\mathcal{D}$ over the state-adversarial uncertainty set $\mathcal{P}_\epsilon^\pi$, there must be an $\alpha \in [0,1]$ such that*

$$\mathbb{E}_{P \sim \mathcal{D}}[J(\pi|P)] = J(\pi|P_\epsilon^{\pi,\alpha}).$$

The proof is in Appendix A.4. It is worth noting that different values of $\alpha$ represent different prior assumptions. For example, $\alpha = 1$ implies that the prior probability of nominal MDP is 1, whereas $\alpha = 0$ indicates that the prior probability of the worst-case MDP is 1. In other words, we can control the value of $\alpha$ to represent different distributions $\mathcal{D}$ and train the policies under various environments. To achieve this goal, we quantify the gap between the *average* performance $\mathbb{E}_{P \sim \mathcal{D}}[J(\tilde{\pi}|P)]$ and *the worst case* performance $J(\tilde{\pi}|P_\epsilon^\pi)$ when updating the current policy $\pi$ to a new policy $\tilde{\pi}$, and then apply an optimization technique to maximize both of them. One naïve bound is as follows.

**Theorem 1** (A naïve connection between the average-case and the worst-case returns). *Given a nominal MDP with state adversaries, when updating the current policy $\pi$ to a new policy $\tilde{\pi}$, the following bound holds (Jiang et al., 2021):*

$$J(\tilde{\pi}|P_\epsilon^\pi) \geq \mathbb{E}_{P \sim \mathcal{D}}[J(\tilde{\pi}|P)] - 2R_{max}\frac{\gamma\mathbb{E}_{P \sim \mathcal{D}}[d_{TV}(P_\epsilon^\pi\|P)]}{(1-\gamma)^2} - 4R_{max}\frac{d_{TV}(\pi,\tilde{\pi})}{(1-\gamma)^2}, \tag{7}$$

*where $R_{max}$ is the maximum reward, $d_{TV}(\pi,\tilde{\pi})$ indicates the total variation divergence between $\pi$ and $\tilde{\pi}$, and $P_\epsilon^\pi$ is the worst state-adversarial transition kernels.*

Theorem 1 indicates that the gap between the *average-* and the *worst-* case performance can be expressed using the MDP shift $\mathbb{E}_{P \sim \mathcal{D}}[d_{TV}(P_\epsilon^\pi\|P)]$ and the policy evolution $d_{TV}(\pi,\tilde{\pi})$. The proof is in Appendix A.5. Note that the bound in Theorem 1 is loose because the value on the right hand side (RHS) of Equation 7 can be tiny. Specifically, the transition kernel probability shift $\mathbb{E}_{P \sim \mathcal{D}}[d_{TV}(P_\epsilon^\pi\|P)]$ is multiplied by the total maximum return $\frac{R_{max}}{1-\gamma}$, and the additional denominator $1-\gamma$ makes the value even smaller since $\gamma$ is usually set to 0.99 in RL applications. As a result, the bound can be meaningless unless the worst-case MDP $P_\epsilon^\pi$ is very close to the average MDP.

Since state perturbation only perturbs states to nearby states, we consider the smoothness of the reward function and transition property to build a tight connection between the average-case and the worst-case returns. Specifically, Lipschitz continuity in reward function has been widely used in the theory of RL (Fehr et al., 2018; Asadi et al., 2018; Ling et al., 2016). The smoothness of the transition kernel also holds in most of the environments (Shen et al., 2020; Lakshmanan et al., 2015). For example, in grid-world, the next state must be adjacent to the current state; and in MuJoCo, the poses of consecutive periods are similar, no matter what the state-action pairs are considered. Formally, we define this smoothness property of transition kernels as:

**Definition 4** ($\delta$-Smooth Transition Kernel in State). *Let $P$ be a transition kernel and $\delta$ be a positive constant. $P$ is a $\delta$-smooth transition kernel in state if*

$$\|s - s'\| \le \delta, \tag{8}$$

*for all $a$ and for all $s, s'$ with $P(s'|s, a) > 0$.*

With the assumption of Lipschitz continuity in reward function and smoothness of transition kernel, we arrive at the following bound:

**Theorem 2** (Connecting Worst-Case and Average-Case Returns). *Given a nominal MDP with two properties: (1) Reward function of corresponding Markov Reward Process (MRP) with respect to any policy is an $L_r$-Lipschitz function. (2) Nominal transition kernel $P_0$ has the smooth transition property $\delta$, where $\|s - s'\|_2 \le \delta, \forall a$ and $\forall P_0(s'|s, a) > 0$. Then, after updating the current policy $\pi$ to a new policy $\tilde{\pi}$, the following bound holds:*

$$J(\tilde{\pi}|P_\epsilon^\pi) \ge J(\tilde{\pi}|P_\epsilon^{\pi,\alpha}) - \frac{4\gamma(\epsilon + \delta)L_r\alpha}{(1 - \gamma)^3} - \frac{4(\gamma(\epsilon + \delta)L_r + (1 - \gamma)^2 R_{max})d_{TV}(\pi, \tilde{\pi})}{(1 - \gamma)^3}, \tag{9}$$

*where $d_{TV}(\pi, \tilde{\pi})$ is total variation divergence between $\pi$ and $\tilde{\pi}$, $P_\epsilon^{\pi,\alpha}$ is a relaxed state-adversarial transition kernel, and $P_\epsilon^\pi$ is a worst-case state-adversarial transition kernel.*

The proof is provided in Appendix A.6. Notably, Theorem 2 holds for any relaxation parameter $\alpha \in [0, 1]$. We now briefly discuss the technical challenges in the proof: (1) *Propagation of state perturbations across time*: The main difficulty lies in the fact that the difference of trajectories under different MDPs would increase in a rather nonlinear and complex manner as time evolves. (2) *Quantifying the difference in rewards among trajectories generated under different transition kernels:* To measure the difference in rewards under different MDPs, it is necessary to consider not only the probability difference at time $t$ but also the difference in rewards at different states. Despite the above challenges, our proof uses the finding that the difference of initial probability of state under two MDPs $P_\epsilon^\pi$ and $P_\epsilon^{\pi,\alpha}$ at time step $t$ can be quantified as $\alpha\Delta_t$, where $0 \le \Delta_t \le 1$. Then under the smoothness conditions of the reward function and the transition matrix, we are able to characterize a tight bound between the average-case and the worst-case performance.

The intuition of Theorem 2 can be expressed using the terms on the RHS of Equation 9. The first term is the average performance of all MDPs in the uncertainty set. The second term penalizes the large value of $\alpha$ because it implies that the relaxed MDP is close to the nominal environment. In other words, we expect the average case performance to be high while pushing the uncertainty set close to the worst-case MDP. Finally, the third term prevents a significant update in a single step by reducing the total variation divergence $d_{TV}(\pi, \tilde{\pi})$.

### 4.4 ONLINE ADAPTATION OF THE RELAXATION PARAMETER

We leverage Theorem 2 to address both the average-case and the worst-case performance. Specifically, we present a *bi-level* approach to maximize the lower-bound of the worst-case performance (i.e., RHS of Theorem 2) since the unknowns $\alpha$ and $\pi$ are correlated. The two tasks are optimized alternatively and iteratively. Details are as follows:

- **Lower-level task for average-case return**: On the lower level, we improve the policy by optimizing the objective $J(\pi_{\theta_t}|P_\epsilon^{\pi_{\theta_{t-1}},\alpha_t})$ under a fixed relaxation parameter $\alpha_t$. This can be done by using any off-the-shelf RL algorithm (e.g., PPO with a clipped objective).

- **Upper-level task for worst-case return**: On the upper level, we design a meta objective $J_{\text{meta}}(\alpha_t)$ to represent the lower bound of the worst case performance (RHS of Equation 9). Hence, the task aims to find a relaxation parameter $\alpha_t$ that can maximize $J_{\text{meta}}(\alpha_t)$. To enable a stable training, we iteratively update $\alpha_t$ by applying the online cross-validation algorithm (Sutton, 1992).

Both the lower and upper level tasks aim to increase the lower bound of the worst-case performance $J(\pi_{\theta_t}|P_\epsilon^{\pi_{\theta_t-1}})$ (Equation 9). In the lower-level, a constant relaxation parameter $\alpha_t$ represents a specific distribution $\mathcal{D}$. It seeks to maximize the average return over all environments in the uncertainty set following distribution $\mathcal{D}$. In the upper-level, the optimization adjusts $\alpha$ to maximize this lower bound. On one hand, increasing $\alpha_t$ improves the average performance $J(\pi_{\theta_t}|P_\epsilon^{\pi_{\theta_t-1},\alpha_t})$ since the average-case moves toward a nominal environment, yet the price is increasing the MDP shift (i.e., the second term of RHS in Equation 9). On the other hand, decreasing $\alpha$ changes the performance and the penalty oppositely. Since $\pi$ is weak initially and its performance gradually improves, the meta objective optimization tends to decrease and then increase $\alpha$ during training.

Algorithm 1 illustrates our implementation. We first update the policy $\pi_{\theta_t}$ to maximize the average-case return $J(\pi_{\theta_t}|P_\epsilon^{\pi_{\theta_t-1},\alpha_t})$ using the proximal policy optimization (PPO). Afterward, we update the relaxation parameter $\alpha$ to ensure that the worst-case return is higher than a specific bound (Equation 9). Note that samples used in the two steps are different (Lines 3 and 6 of Algorithm 1) because the meta objective optimization is an online method. In addition, we chose PPO as a base algorithm since it prevents the model from being updated significantly in a single step. It helps to control the penalty term $d_{TV}(\pi, \tilde{\pi})$ in Theorem 2. The implementation details are provided in Appendix A.7.

---

**Algorithm 1:** Relaxed State-Adversarial Policy Optimization

**Input :** MDP $(\mathcal{S}, \mathcal{A}, P_0, r, \gamma)$, Objective function $L$, step size parameter $\eta$, number of iterations $T$, $P_0$ is the nominal transition kernel, $\epsilon$-Neighborhood

1  Initialize the policy $\pi_{\theta_0}$ **for** $t = 0, \ldots, T-1$ **do**

2      Sample the tuple $\{s_i, a_i, r_i, s_i'\}_{i=1}^{T_{\text{upd}}}$, where $a_i' \sim \pi_{\theta_t}(\cdot|s_i')$, and $s_i' \sim P_0(\cdot|s_i, a_i)$

3      Evaluate $J(\pi_{\theta_t}|P_\epsilon^{\pi_{\theta_t-1},\alpha_t})$

4      Update the policy to $\pi_{\theta_{t+1}}$ by applying multi-step SGD to the objective function as PPO

5      Sample the tuple $\{s_i, a_i, r_i, s_i'\}_{i=1}^{T'_{\text{upd}}}$, where $a_i' \sim \pi_{\theta_{t+1}}(\cdot|s_i')$, and $s_i' \sim P_0(\cdot|s_i, a_i)$

6      Update the relaxation parameter to $\alpha_{t+1}$ via one SGD update with respect to the meta-objective

7  **end**

---

## 5    EXPERIMENTAL RESULTS AND EVALUATIONS

We conducted two experiments on Mujoco (Todorov et al., 2012) to evaluate the performance of our relaxed state adversarial policy optimization (RAPPO). All the baselines and our method were implemented on the PPO (Schulman et al., 2017), and the default training parameters were used. In addition, the results were averaged from five different runs/seeds.

**Robustness against Environmental Adversaries.** We compared our RAPPO with the latest DR method, MRPO (Jiang et al., 2021), to evaluate its robustness against the uncertainty of environmental parameters[1]. Agents trained using the two methods were evaluated in the environments, in which the size and gravity were drifted in the range of 0.6 - 1.4. To simulate the situation that domain knowledge is unavailable, during training, MRPO perturbed mass and friction in the range of 0.8 - 1.2, and our RAPPO attacked the states by its value function. Figure 2 shows the subtractions of the rewards of the two methods. As can be seen, our RAPPO outperformed MRPO since state adversaries were more general than environmental adversaries. Agents trained by MRPO could perform poorly when the perturbations in the training and testing environments were different.

**Robustness Against States Adversaries.** We compared our RAPPO with SCPPO (Kuang et al., 2021) to evaluate its robustness against state adversaries. Both of the methods perturb states to improve agents' robustness. We also included vanilla PPO in the experiment because it is the base algorithm of RAPPO and SCPPO. To achieve a fair comparison, the parameters used in RAPPO and SCPPO were the same. Specifically, we set $\epsilon$ to 0.015, 0.002, 0.03, 0.001, and 0.005 to the environments of HalfCheetah-v2, Hopper-v2, Ant-v2, Walker-v2, and Humanoid2d-v2, respectively. The parameters were chosen according to the variance of actions in the environments.

---

[1]We obtained the official implementation of MRPO from http://proceedings.mlr.press/v139/jiang21c.html and used their default parameter setting.

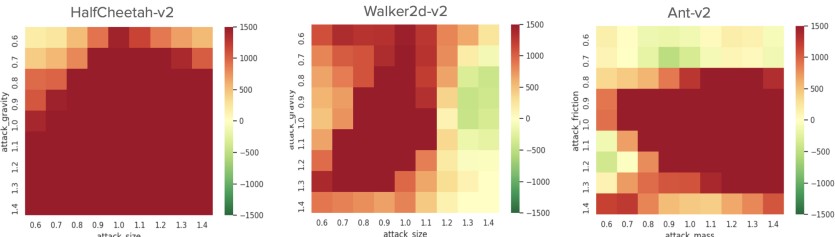

Figure 2: We perturbed the size and gravity of the environments and measured the mean rewards achieved by the agents trained using a DR method, MRPO, and our RAPPO. The heatmaps show the subtractions of MRPO's reward from RAPPO's reward. The higher value (red) indicates that RAPPO outperformed MRPO.

| Environment | | Nominal | $\sigma = 0.005$ | $\sigma = 0.01$ | $\sigma = 0.015$ | $\sigma = 0.02$ | $\sigma = 0.025$ |
|---|---|---|---|---|---|---|---|
| HalfCheetah | PPO | $5286 \pm 1004$ | $4280 \pm 1552$ | $3186 \pm 1875$ | $1996 \pm 1743$ | $1256 \pm 1251$ | $819 \pm 1003$ |
| | SCPPO | $6157 \pm 709$ | $5046 \pm 1533$ | $3367 \pm 2090$ | $1795 \pm 1758$ | $875 6\pm 1259$ | $60 \pm 791$ |
| | RAPPO-C | $5830 \pm 779$ | $5185 \pm 782$ | $4084 \pm 1266$ | $2743 \pm 1644$ | $1459 \pm 1439$ | $406 \pm 912$ |
| | RAPPO | $6146 \pm 742$ | $\mathbf{5519 \pm 774}$ | $\mathbf{4353 \pm 1510}$ | $\mathbf{3087 \pm 1568}$ | $\mathbf{1878 \pm 1287}$ | $\mathbf{846 \pm 951}$ |
| | | Nominal | $\sigma = 0.0008$ | $\sigma = 0.0016$ | $\sigma = 0.002$ | $\sigma = 0.0024$ | $\sigma = 0.003$ |
| Hopper | PPO | $3330 \pm 619$ | $1357 \pm 787$ | $615 \pm 194$ | $494 \pm 151$ | $462 \pm 141$ | $417 \pm 131$ |
| | SCPPO | $2644 \pm 951$ | $1369 \pm 620$ | $876 \pm 347$ | $773 \pm 357$ | $782 \pm 437$ | $732 \pm 412$ |
| | RAPPO-C | $2497 \pm 1041$ | $1626 \pm 1064$ | $1122 \pm 817$ | $795 \pm 426$ | $704 \pm 393$ | $521 \pm 201$ |
| | RAPPO | $3301 \pm 520$ | $\mathbf{2198 \pm 859}$ | $\mathbf{1457 \pm 537}$ | $\mathbf{1244 \pm 584}$ | $\mathbf{1067 \pm 605}$ | $\mathbf{1014 \pm 779}$ |
| | | Nominal | $\sigma = 0.001$ | $\sigma = 0.0015$ | $\sigma = 0.002$ | $\sigma = 0.0025$ | $\sigma = 0.003$ |
| Walker2d | PPO | $3781 \pm 1165$ | $1564 \pm 1285$ | $903 \pm 52$ | $763 \pm 353$ | $628 \pm 241$ | $575 \pm 222$ |
| | SCPPO | $4313 \pm 979$ | $2647 \pm 1584$ | $1604 \pm 1082$ | $985 \pm 704$ | $772 \pm 492$ | $666 \pm 412$ |
| | RAPPO-C | $4113 \pm 899$ | $2394 \pm 1471$ | $1881 \pm 1398$ | $1520 \pm 1387$ | $1249 \pm 1282$ | $888 \pm 970$ |
| | RAPPO | $4608 \pm 962$ | $\mathbf{3998 \pm 1487}$ | $\mathbf{3298 \pm 1478}$ | $\mathbf{2160 \pm 1408}$ | $\mathbf{1470 \pm 1013}$ | $\mathbf{1173 \pm 783}$ |
| | | Nominal | $\sigma = 0.01$ | $\sigma = 0.02$ | $\sigma = 0.03$ | $\sigma = 0.04$ | $\sigma = 0.05$ |
| Ant | PPO | $6075 \pm 889$ | $\mathbf{4489 \pm 1342}$ | $2071 \pm 1156$ | $1016 \pm 523$ | $703 \pm 283$ | $615 \pm 248$ |
| | SCPPO | $5915 \pm 728$ | $4203 \pm 1441$ | $1661 \pm 951$ | $831 \pm 398$ | $609 \pm 320$ | $489 \pm 273$ |
| | RAPPO-C | $5954 \pm 746$ | $4380 \pm 1299$ | $2077 \pm 1117$ | $1010 \pm 568$ | $668 \pm 282$ | $537 \pm 225$ |
| | RAPPO | $6022 \pm 698$ | $4381 \pm 1357$ | $\mathbf{2284 \pm 1225}$ | $\mathbf{1038 \pm 553}$ | $\mathbf{733 \pm 255}$ | $\mathbf{672 \pm 219}$ |
| | | Nominal | $\sigma = 0.003$ | $\sigma = 0.004$ | $\sigma = 0.005$ | $\sigma = 0.006$ | $\sigma = 0.007$ |
| Humanoid | PPO | $5357 \pm 1618$ | $3033 \pm 1834$ | $2373 \pm 1742$ | $1802 \pm 1446$ | $1287 \pm 1068$ | $939 \pm 750$ |
| | SCPPO | $5410 \pm 1340$ | $3196 \pm 1781$ | $2387 \pm 1472$ | $1783 \pm 1256$ | $1271 \pm 838$ | $1060 \pm 678$ |
| | RAPPO-C | $5169 \pm 1468$ | $3031 \pm 1810$ | $1941 \pm 1336$ | $1550 \pm 1165$ | $1035 \pm 551$ | $874 \pm 458$ |
| | RAPPO | $5355 \pm 1491$ | $\mathbf{3768 \pm 1972}$ | $\mathbf{3227 \pm 1883}$ | $\mathbf{2537 \pm 1698}$ | $\mathbf{1747 \pm 1274}$ | $\mathbf{1350 \pm 1133}$ |

Table 1: We compared the performance of agents trained using PPO, SCPPO, our RAPPO-C (i.e., constant relaxation parameter $\alpha$) and RAPPO in Mujoco environments under multiple degrees of state perturbation. Mean and Standard deviations are reported.

Table 1 shows the testing results. We attacked the agents using their respective value functions under multiple strengths. Specifically, we repeated the experiments from 5 different seeds and generated 50 trajectories for each seed from different initial states for evaluation. The means and standard deviations of the rewards were reported. Clearly, the results fulfilled Lemma 1, where agents' performance decreased as the strength of attack increased. In addition, our RAPPO was competitive to PPO and SCPPO in nominal environments, and its performance decreased the slowest as the strength of attack increased. It deserves noting that the attacks in the last two columns of Table 1 were stronger than that of the worst-case. Our RAPPO performed the best in the environments.

**Extending SAPPO Using Relaxed State Adversaries.** While our RAPPO successfully improves the robustness of agents against state adversaries, a classical method, SAPPO (Zhang et al., 2020), can help agents against the perturbation of state observations. We thus extended SAPPO by adopting our relaxed state adversarial attacks during training and evaluated its effectiveness. Similarly, we compared the methods on the trajectories of 5 seeds and 50 initial states. Table 2 shows the results. As indicated, the extended RA_SAPPO outperformed SAPPO in most of the environments, particularly under strong attacks.

**Steady Improvements of the Average and Worst Case Environments.** We apply a *bi-level* approach to optimize the average and worst-case environments during training. To verify the feasibility of this approach, we evaluated the agents' performance under these two cases during training. To determine the worst-case result, we generated 50 trajectories from different initial states,

| Environment | | Nominal | $\sigma = 0.005$ | $\sigma = 0.01$ | $\sigma = 0.015$ | $\sigma = 0.02$ | $\sigma = 0.025$ |
|---|---|---|---|---|---|---|---|
| Halfcheetah | SAPPO | 4928 ± 370 | 4765 ± 359 | 4485 ±394 | 4036 ± 582 | 3282 ± 1175 | 2533 ± 1495 |
| | RA_SAPPO | 5784 ± 1081 | **5371 ± 1323** | **4874 ± 1311** | **4775 ± 933** | **4106 ± 1273** | **3152 ± 1750** |
| | | Nominal | $\sigma = 0.001$ | $\sigma = 0.0015$ | $\sigma = 0.002$ | $\sigma = 0.0025$ | $\sigma = 0.003$ |
| Walker | SAPPO | 4135 ± 962 | 2211 ± 1322 | 940 ± 405 | 673 ± 318 | 667 ± 326 | 614 ± 311 |
| | RA_SAPPO | 4539 ± 1014 | **3229 ± 1590** | **1564 ± 1410** | **921 ± 789** | **832 ± 806** | **746 ± 772** |
| | | Nominal | $\sigma = 0.003$ | $\sigma = 0.004$ | $\sigma = 0.005$ | $\sigma = 0.006$ | $\sigma = 0.007$ |
| Humanoid | SAPPO | 5736 ± 1194 | 3690 ± 2068 | 2926 ± 1956 | 1944 ± 1438 | 1409 ± 1098 | 1156 ± 789 |
| | RA_SAPPO | 5320 ± 1164 | **3960 ± 2082** | **3335 ± 2117** | **2882 ± 2066** | **2129 ± 1776** | **1567 ± 1474** |

Table 2: We extended the SAPPO by adopting the relaxed state adversarial strategy and evaluated whether the extension (i.e., RA_SAPPO) can improve the agents' robustness against state perturbation. Mean and Standard deviations are reported.

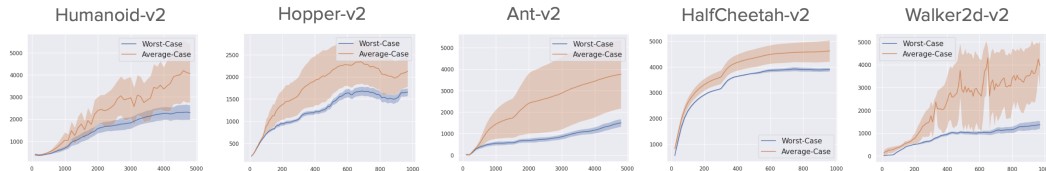

Figure 3: Our RAPPO can steadily improve the average-case and worst-case rewards during training. The solid lines and shaded areas indicate the mean and standard deviation of the rewards, respectively. Note that the variance of the average-case rewards is caused by different adversarial strengths.

perturbed states with the same strength as the training $\epsilon$, and then averaged the rewards. In contrast, the average-case result was determined from 50 initial states and 10 different perturbation strengths, which were uniformly distributed between 0 and $\epsilon$. In total, the rewards of $50 \times 10$ trajectories were averaged. Figure 3 shows that our RAPPO can steadily improve the average-case performance without sacrificing the worst-case performance. Note that the high variance of the average-case rewards is reasonable because of different adversarial strengths.

**The value of the relaxation parameter $\alpha$.** Our meta-objective optimization determines the relaxation parameter $\alpha$ (Equation 6) to control the strengths of state adversaries during training. While $\alpha$ is unknown, an intuitive idea is to consider $\alpha$ a hyper-parameter and let users specify the value. However, we point out that the value of $\alpha$ should vary at different training stages since agents are weak initially and can perform well after training. To verify that a dynamic $\alpha$ is over a constant $\alpha$ (i.e., RAPPO-C), we evaluated the performance of agents under state perturbed environments. In the experiments, we set $\alpha = 0.5$ for RAPPO-C since it is in the middle of nominal and worst-case environments. The remaining parameters between the methods were exactly the same. As indicated in Table 1, RAPPO outperformed RAPPO-C without a doubt. We also refer readers to Appendix A.8 for the dynamics of $\alpha$ during training.

## 6    CONCLUSIONS

We have presented a relaxed state adversarial policy optimization to improve the robustness of agents against the uncertainty of environments. Compared to the methods in DR, we perturbed states using the adversarial attack so as to decouple randomization from simulators. Neither prior knowledge of selecting environmental parameters nor prior assumption of parameter distribution are needed. In addition, we introduced a relaxation strategy to tackle the over-conservative problem caused by state adversarial attacks. Our policy optimization maximizes rewards in the average-case while holding the lower-bound rewards in the worst-case environments simultaneously. Experiment results and theoretical proofs demonstrate the effectiveness of our method.

**Limitations and Future Work.**  Our relaxation method is state-independent, in which the value of $\alpha$ is adjusted according to the overall performance of policy. Since the degrees of difficulty vary from states to states, it will be interesting to investigate the state-dependent relaxation method. In addition, we currently assume that each dimension of states is equally important, which may not be the case. We will also explore the weight of each dimension when perturbing states in the future.

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
