# OpenReview forum: "Revisiting Domain Randomization Via Relaxed State-Adversarial Policy Optimization"
_ICLR.cc/2023/Conference — Submitted to ICLR 2023_

### Official Review · Reviewer_1qFW · 2022-10-24

**Confidence:** 4
**Correctness:** 3
**Technical Novelty And Significance:** 3
**Empirical Novelty And Significance:** 3
**Recommendation:** 6

**Clarity, Quality, Novelty And Reproducibility:**

The paper is well written and clear. While aspects of this work exist in prior work, the authors present a novel algorithm and compare against prior related methods empirically and show connections to prior work theoretically.

**Strength And Weaknesses:**

Strengths
- Very clearly written and well motivated, was a pleasure to read.
- Motivation behind the method and theoretical insights seem well constructed.

Weaknesses
- The high variance on the experimental results raise some questions: within the error bounds, RAPPO is often comparable with the baselines (e.g. for Humanoid and Ant). To make the improvements more clear, I would recommend either running more experiments and/or only bolding the results that are significantly the best performing.

Questions
- Have the authors considered other schedules for a dynamic alpha? The authors compare against a constant alpha, but I would be curious if a linearly increasing schedule is also sufficient and maybe avoids some of the complexities of learning alpha. It looks like from Figure 5 in the Appendix that alpha can vary quite greatly across seeds, suggesting that it may be difficult to learn. It would also be interesting to see how the different alpha dynamics corresponds to different policy performance.

Additional Feedback / Minor Notes
- Formatting in $V_P^\pi$ definition at the bottom of page 3 — sum over $a_t$ instead of $at$
- Use of $\epsilon$ for FGSM in Remark 1 is slightly confusing with use of $\epsilon$ as perturbation radius in Remark 2, maybe switch to a different symbol
- 'Upper-lever' instead of upper-level in 4th line on page 7
- Figure 2 caption — last sentence remove ‘the more’
- State meta-objective in algorithm for additional clarity

**Summary Of The Paper:**

To overcome some of the limitations and assumptions required for domain randomization (knowledge of underlying simulator parameters and parameter distributions), the authors tackle the problem of robust RL through adversarial state perturbations instead. However, rather than only accounting for the worst-case perturbations, which can lead to overly conservative agents, the proposed method ‘Relaxed State-Adversarial Algorithm’ simultaneously maximizes both the worst-case and average-case policy performance with the hope that additionally optimizing for average-case performance will prevent overly-conservative behaviours. The authors present a theoretical derivation of their method and evaluate it empirically against relevant prior work.

**Summary Of The Review:**

I recommend a 6. The paper is well motivated, the method presented is straightforward and intuitive, and supported by both theoretical proofs and empirical results comparing against SOTA methods. However, the paper would be strengthened if the empirical results were more clearly statistically significant — right now the standard deviations on the results are quite large, which makes it difficult to draw confident conclusions from the experiments.

---

### Official Review · Reviewer_PgB5 · 2022-10-24

**Confidence:** 3
**Correctness:** 3
**Technical Novelty And Significance:** 3
**Empirical Novelty And Significance:** 4
**Recommendation:** 6

**Clarity, Quality, Novelty And Reproducibility:**

The paper has good clarity except for a few details.

The proposed method is novel.

It’s reproducible.


**Strength And Weaknesses:**

Strength:

The paper has good writing and clear delivery of the idea. The theoretical support of the proposed method seems to be solid. The proposed algorithm is novel and interesting. Experiments support the effectiveness of the method well.

Weakness:

What’s $J_{meta}$? I checked the appendix and it seems the algorithm uses a learned approximation of the $J_{meta}$, please explain and justify this in the main paper.

Is state adversary really necessary? I agree that the state adversary is more general than other types of environment adversary like parameter adversary, etc. However, is state adversary really capturing the difference between the simulation and reality? Given that the state adversary in the paper is independently generated for each state, there is no consecutive dependency along different steps, while I guess most dynamics mismatch between simulation and reality preserve this dependency.

Confusion about Lemma 2. It is true in Lemma 2 that a certain $\alpha$ exists to make the relaxed state-adversarial equal to the expectation of uncertainty set in terms of objective value. But when using the relaxed state-adversarial in the algorithm, the relaxed state-adversarial is actually not corresponding to the average-case return, right? Because (1) the $\alpha$ is changing and (2) the true $\alpha$ satisfying Lemma 2 is unknown. So why optimizing over the relaxed state-adversarial corresponds to optimizing over the average-case return?

From the results in the paper, the $\alpha$ is decreasing first and increasing later, which indicates the algorithm is optimizing over the average-case objective more at first and later more on the worst-case objective. And the optimization over the worst-case objective is achieved essentially through shrinking the uncertainty set ($\alpha$ gets larger), but not necessarily improving the policy performance on the full uncertainty set (corresponding to $\alpha=1$), so how does this really help with the policy facing the worst case?

For experiments, although the RAPPO outperforms SCPPO, it may not be a fair comparison since RAPPO is designed with state adversary while SCPPO is not, but both of them are evaluated under state adversary.


**Summary Of The Paper:**

The paper proposes a method to mitigate the problem of over-conservative policy when optimizing under the worst-case state adversary. Under certain conditions, it proves a tighter bound of the average-case return and worst-case return under state adversary. A two-level update algorithm is proposed to automatically tune the optimization on the worst-case and average-case objective. Experiments on MuJoCo show the learned policies to be more robust against different levels of state adversary.

**Summary Of The Review:**

Overall, this is an interesting paper with a novel algorithm to solve the robustness of RL agent against adversary. Several questions need to be answered well in the paragraph to make it better. Experimental results are strong. Perhaps more ablation studies would be good. Aslo it could be better to test the algorithms on some other task suites.

---

### Official Review · Reviewer_E4mE · 2022-10-25

**Confidence:** 4
**Correctness:** 2
**Technical Novelty And Significance:** 2
**Empirical Novelty And Significance:** 2
**Recommendation:** 5

**Clarity, Quality, Novelty And Reproducibility:**

The presentation is unclear in a few places, for example:
- The definition of MDP in this paper is redefined to let P be a set of transition functions rather than a fixed transition function.  It is weird to call this an MDP, since it is not, since it is closer to a Robust MDP
- I don't know what is meant by the phrase "it stands for the average performance of the uncertainty set"
- it guarantees the improvement of the worst-case MDP.  What is the "worst case MDP" and what would it mean to "improve" it?

**Strength And Weaknesses:**

Strengths:
* The conservatism (If I've understood it) that they identified can be an bottleneck to performance in some settings
* The approach is relatively simple
* The theoretical results appear correct

Weaknesses:
* It is unclear what is meant exactly by over- conservatism

A) Could mean a specific sort of feed-back effect between the TD updates and the Q-Values as shown in Figure 1.  (that is, the policy training doesn't find the policy the designer intended)

B) Could mean that it is "conservative" to a point that it is bad.  (that is, the policy training worked but conservative policy the designer intended to find was too conservative so does not perform well)

Other than Figure 1, the rest of the paper can be viewed as B, and it seems that A is not mentioned again apart from Figure 1.  If A is the point, then I would want to see empirical evidence that this happens by default in the experimental domains.

* Figure 1 is difficult to understand and needs a much more mechanistic explanation if it going to be a core part of the argument.  Once I understood the claim, it felt like I had to rederive the result on my own rather than it being explained.  It is not initially clear how errors would form in the first place, or why those errors would get bigger over time.  Even now, it is not clear why the worst point would be exempt from those errors making its valuation even worse.

* The evaluation is empirically weak
 The policy was only compared to other policies which were not robustified along all of the dimensions their method was able to be made robust to.  This is justified by saying that the other methods would require instrumentation of the simulator in order to modify all dimensions, but it seems that their method would also require a slightly different sort of instrumented simulator, and that the other methods could be trivially modified to use that sort of instrumentation.

 More specifically, it seems that the authors want to disallow things that modify the transition function, but allow for modifications of the state.  Since many simulators don't have a reason to allow arbitrary state modifications, it is an awkward line to draw.  But given that we are drawing that line, it would make the most sense to benchmark against methods which can also modify the state in the same way.  Since adding a perturbation to the state is a special case of modifying the transition function, all the other considered methods should already work in this setting, if configured properly.  Thus allowing the other methods to make state perturbations would be the appropriate baseline

The results, as given in Table 1 are very difficult to interpret given the highlighting.  Results are highlighted as ``best'' which are not statistically distinguishable from the others, and so it makes the results at a glance look more impressive than they are.

* The theoretical results are straightforward.  While they are nice to have, and good due diligence, I would not consider them as grounds for acceptance on their own.


**Summary Of The Paper:**

This paper identifies an issue they call "over-conservatism" with adversarial training methods, which causes more conservative behavior than is desirable.  This is adressed by instead optimizing for a mixture of average case and worst case performance.

**Summary Of The Review:**

I am recommending rejection since the core argument of the paper is unclear, and the empirical results do not sufficiently support that argument.

---

### Official Review · Reviewer_H7Ak · 2022-10-25

**Confidence:** 4
**Correctness:** 2
**Technical Novelty And Significance:** 3
**Empirical Novelty And Significance:** 3
**Recommendation:** 5

**Clarity, Quality, Novelty And Reproducibility:**

The paper is easy to follow with proper explanations in general. There are some inconsistent capitalization mixing $P^{\pi}_\epsilon$ and $p^{\pi}_\epsilon$. In the proof of Theorem 1 in A.5, the notation for the reward function is incorrect.

**Strength And Weaknesses:**

Strength:
- The connection of relaxed state-adversarial MDPs with domain randomization is very interesting, and it provides an average-case robustness metric which might help mitigate over-conservative in adversarial trainings.

- Numerical evaluations in Mujoco tasks show that the proposed method outperforms prior methods in cases with perturbation of state observations as well as cases with adversarial attacks on their value functions.

Weaknesses:
- The proofs of Theorem 1 and 2 may be correct, but the bounds are a bit strange and they may not correspond the desired performance of the algorithm. In particular, the current policy $\pi$ does not appear in both the LHS of (7) and (9). This means that one can set $\tilde{\pi} = \pi$ in (7) and (9) and obtain tighter bounds without their last terms. This lacks of dependence on the current policy is likely different from what has been implemented in the algorithm. One can also observe that, although Theorem 1 is mostly the same as Theorem 1 in Jiang et al., 2021, the LHS of the associated equation in Jiang et al., 2021 corresponds to the adversary for the current policy $\pi$, while the LHS of (7) only corresponds to the adversary of the new policy $\tilde{\pi}$.

- No discussion and/or numerical evaluations on how good the lower bound of Theorem 2 is. As mentioned in the previous point, the last term in (9) is likely redundant by setting $\tilde{\pi} = \pi$. The tightness of the bounds is questionable.

- It's not clear if step of minimizing the lower bound of the worst-case return really contributes to better performance. What if one just fixed $\alpha$ without the upper-level task?

**Summary Of The Paper:**

The paper proposed an adversarial training approach for robust RL. The approach is a bi-level method where the lower-level task minimizes  the average-case return against state adversarial attacks while the upper-level task handle the worst-case adversarial return via minimizing a lower bound. Empirical experiments show improved performance from baselines on Mujoco tasks with environmental perturbations and state adversarial attacks.

**Summary Of The Review:**

The idea of to train in the relaxed state-adversarial MDP is interesting and seems to be a promising approach for RL robustness, but the paper might have some technical issues, and more evaluations may be needed to identify why and how the proposed method actually provides performance improvement against adversaries.

---

### Decision · Program_Chairs · 2023-01-20

**Decision:**

Reject

**Justification For Why Not Higher Score:**

Several concerns on the theoretical justification and empirical evaluation exist in this work.

**Justification For Why Not Lower Score:**

N/A

**Metareview: Summary, Strengths And Weaknesses:**

The paper presents a new adversarial approach for training robust RL policies. Intuitively, the approach tries to maximize the returns when faced with adversarial state attacks. Experiments are shown on Mujoco, which highlight improvements over prior works like domain randomization. The technical ideas have also been appreciated by the reviewers. However, there still remains significant concerns with this paper. Several reviewers have raised questions about the correctness and relevance of the theroretical results in this work. Furthermore, the empirical results do not seem move the state-of-the-art in robust RL. Minor questions about the statistical significance of the results have also been raised, which still remains after the rebuttal.

I would like to add that we appreciate the significant time and effort taken by the authors in replying to the reviewers. There are interesting ideas in this work, and I hence recommend the authors to resubmit a revised version of this work with the additional discussions.